# Glutathione S-Transferase Gene Polymorphisms as Predictors of Methotrexate Efficacy in Juvenile Idiopathic Arthritis

**DOI:** 10.3390/biomedicines12081642

**Published:** 2024-07-24

**Authors:** Sanda Huljev Frkovic, Marija Jelusic, Kristina Crkvenac Gornik, Dunja Rogic, Marijan Frkovic

**Affiliations:** 1Department of Paediatrics, University Hospital Centre Zagreb, University of Zagreb School of Medicine, 10000 Zagreb, Croatia; sanda.huljev@gmail.com (S.H.F.); marija.jelusic@mef.hr (M.J.); 2Department of Laboratory Diagnostics, University Hospital Centre Zagreb, 10000 Zagreb, Croatia; kristina.crkvenac@kbc-zagreb.hr (K.C.G.); dunjarogic@hotmail.com (D.R.)

**Keywords:** juvenile idiopathic arthritis, biomarker, glutathione S-transferase, methotrexate, efficacy

## Abstract

Because of the unpredictable efficacy of methotrexate (MTX) in the treatment of juvenile idiopathic arthritis (JIA), the possibility of a favourable outcome is reduced in more than 30% of patients. To investigate the possible influence of glutathione S-transferase M1 (GSTM1) and T1 (GSTT1) gene deletion polymorphisms on MTX efficacy in patients with JIA, we determined these polymorphisms in 63 patients with JIA who did not achieve remission and 46 patients with JIA who achieved remission during MTX therapy. No significant differences were observed in the distribution of single GSTM1 or GSTT1 deletion polymorphisms or their combination between the two groups: 58.7% to 63.5%; *p* = 0.567, 17.4% to 22.2%; *p* = 0.502, and 13% to 12.7%; *p* = 0.966, respectively. Our results suggest that GSTM1 and GSTT1 deletion polymorphisms do not influence the efficacy of MTX in patients with JIA. Additional studies are required to determine the possible influence of GST deletion polymorphisms on MTX efficacy in patients with JIA.

## 1. Introduction

Juvenile idiopathic arthritis (JIA) is the most common among chronic rheumatic diseases of childhood [1,2], with an incidence of 2–20/100,000 and a prevalence of 16–150/100,000 annually [3,4]. It comprises a heterogeneous group of arthritis types of unknown origin and is caused by a complex interaction between non-Mendelian genetic and environmental factors [5].

Notable rheumatology societies and working groups recommend a stepwise therapy approach, incorporating non-steroidal anti-inflammatory drugs (NSAIDs), glucocorticoids, and conventional disease-modifying anti-rheumatic drugs (cDMARDs), such as methotrexate (MTX), biologic drugs (bDMARDs), and physical/psychosocial therapies, for managing JIA while considering the unique traits of each disease type [6,7,8,9].

Contemporary JIA treatment goals aim to achieve disease remission according to Wallace’s criteria [10,11,12]. Early control of the inflammatory process, i.e., achieving disease control during the “window of opportunity”, correlates with more favourable treatment outcomes [13,14].

The first drug of the second line in JIA treatment is MTX, a slow-acting cDMARD. The response to MTX in patients with JIA is heterogeneous, with more than 30% of patients exhibiting no effect of MTX on remission. Because continuous administration of MTX for at least 3–6 months is necessary to evaluate its effect in JIA patients, the probability of early remission and favourable treatment outcomes is reduced in unpredictable cases. Therefore, identification of a reliable marker for the early exclusion of JIA patients unlikely to benefit from MTX who require third-line therapy, i.e., bDMARDs, is crucial [15,16,17,18].

Drug intolerance, an alternative reason for excluding MTX from treatment regimens necessitating the use of other cDMARDs or bDMARDs without MTX coadministration, was not the focus of this study [8].

Glutathione S-transferases (GSTs) are a group of widely distributed, multifunctional enzymes that catalyse the conjugation of toxic reactive components with glutathione (GSH) in prokaryotic and eukaryotic cells. It facilitates the detoxification of many exogenous and endogenous metabolites as well as their inactivation and elimination from cells [19,20]. The enzymatic activity of GSTs depends on deletion polymorphisms or single-nucleotide polymorphisms (SNPs) of GST genes, the amount of exogenous or endogenous substrates that modulate GST gene expression, the stable supply of GSH under the action of gamma-glutamylcysteine and glutathione synthetases, and the activity of specific transport molecules that remove GSH conjugates from the cell [20,21,22]. Polymorphisms of the genes encoding cytosolic GSTs have been described in different classes, with the most attention being paid to alleles of the classes mi (M), theta (T), and pi (P).

Studies have shown that MTX induces an increase in enzymatic GST activity in experimental animals [23]. The lack of activity of GSTs in deletion polymorphisms is associated with a reduced ability to eliminate cyclophosphamide and its toxic metabolites in the treatment of lupus nephritis; therefore, these “null” polymorphisms are linked to an enhanced therapeutic response [24]. Because GSTs (GSTM1, GSTP1, and GSTT1) metabolise the original drug or metabolites of steroids, vincristine, anthracycline, methotrexate, cyclophosphamide, and etoposide, correlations between clinical response and GST polymorphisms have also been investigated in different therapeutic modalities, including MTX-containing protocols in patients with SLE, IBD [25,26,27], and malignant diseases, especially leukaemias [28,29,30]. On the other hand, malignant cells show pronounced GST activity in some malignant diseases, leading to resistance to certain chemotherapeutics because of their accelerated elimination [31,32,33].

This study aimed to explore the influence of GSTM1 and GSTT1 polymorphisms on MTX efficacy, assuming that the deletion polymorphisms leading to reduced enzymatic activity would cause MTX accumulation and subsequently enhance its effectiveness in patients with JIA. Therefore, the determination of GSTM1 and GSTT1 polymorphisms would be beneficial for predicting the effectiveness of MTX.

## 2. Materials and Methods

### 2.1. Study Design and Population

This prospective cohort study was conducted among patients diagnosed with JIA between 1 July 2016 and 30 June 2022 at the Division for Clinical Immunology and Rheumatology, Department of Paediatrics, University Hospital Centre Zagreb.

The study included JIA patients treated with methotrexate (15 mg/m^2^, peroral or subcutaneous formulation) as monotherapy for at least three months. The study excluded JIA patients with comorbidities or overlapping syndromes, patients who were not prescribed MTX, and those whose MTX treatment had been altered because of intolerance or non-adherence.

JIA (including the type of disease) was diagnosed in all patients according to the International League of Associations for Rheumatology (ILAR) criteria [1,2], whereas clinical remission was determined according to the Wallace criteria [10,11,12]. All patients participating in the study were treated in accordance with the American College of Rheumatology (ACR) guidelines for the treatment of JIA or revised ACR guidelines for the treatment of systemic JIA [6,7].

The data on initial symptoms, laboratory findings, disease course, and treatment were systematically recorded.

Patients were divided into two groups according to their MTX response. The first group comprised patients with JIA in whom stable clinical remission lasting for at least 6 months was achieved during MTX therapy (“MTX” group), and the second group comprised patients with JIA in whom stable clinical remission was not achieved during MTX treatment (non-responders to MTX), i.e., patients in whom stable clinical remission was achieved by adding bDMARD to MTX or replacing MTX with bDMARD (“MTX + bDMARD” group).

Finally, GSTM1 and GSTT1 deletion polymorphisms were determined in all study participants.

### 2.2. Ethical Statement

The study was conducted in accordance with the Declaration of Helsinki and approved by the Ethics Committee of UHC Zagreb (class: 8.1-16/94-2, No: 02/21 AG, date: 7 June 2016).

Written informed consent to participate in the study (including DNA analysis) was obtained from all parents/legal guardians and minor participants aged 12 years or older.

### 2.3. Laboratory Analyses

DNA was isolated from peripheral blood using a commercial set for DNA isolation, followed by GSTM1 and GSTT1 genotyping using polymerase chain reaction (PCR) [34]. Genomic DNA was isolated from 2 mL of peripheral venous blood using a commercial DNA isolation kit according to the manufacturer’s protocol (Nucleospin, Macherey Nagel, Duren, Germany, Cat. No. NC1105387). The concentration of DNA obtained using this procedure was 20–60 ng/µL. The isolated DNA concentration was measured using a fluorometer (Invitrogen Qubit 2.0 Fluorometer, Thermo Fisher Scientific, Waltham, MA, USA).

We used the Qubit dsDNA HS Assay kit (Thermo Fisher Scientific, Waltham, MA, USA, Cat. No. Q32851), which is suitable for all Qubit devices. The purity of DNA isolated from all subjects was determined using a NanoDrop 2000 spectrophotometer (Thermo Fisher Scientific, Waltham, MA, USA) from 1 µL of DNA isolate based on the wavelength ratio of DNA/proteins (260/280 nm), which is approximately 1.8, and the ratio of DNA/salt (260/230 nm), which is 2.0–2.2. GSTM1 and GSTT1 gene polymorphisms were determined using PCR.

Exon 7 of cytochrome P450 family 1 subfamily A member 1 (CYP1A1) was used as an internal control for PCR success [34].

The total volume of the PCR reaction was 25 µL (19.85 µL H20; 0.5 µL primers 5′ and 3′ concentration 10 pmol/µL; 2.5 µL buffer 10X containing 25 mM MgCl2 (AmpliTaq DNA Polymerase with Buffer II, Applied Biosystems, Thermo Fisher Scientific, Waltham, MA, USA, Cat. No. N8080161); 0.5 µL deoxyribonucleoside–triphosphate (dNTP) concentration 10 mM (Thermo Fisher Scientific, Waltham, MA, USA, Cat. No. R0193); 0.15 µL polymerase Taq concentration 5 U/µL (AmpliTaq DNA Polymerase with Buffer II, Applied Biosystems, Thermo Fisher Scientific, Waltham, MA, USA, Cat. No. N8080161); 1 µL DNA concentration 50–70 ng/µL).

The PCR products of GSTM1 and GSTT1 genes were analysed using 2% agarose gel.

Agarose gel electrophoresis was performed for 20 min at a voltage of 200 V on a Wide Mini-Sub Cell GT Horizontal Electrophoresis System, 15 × 10 cm tray, with PowerPac Basic Power Supply (BIORAD, Hercules, CA, USA).

Separated DNA fragments fluoresce under UV light from an ImageQuant100 transilluminator (Life Sciences, Uppsala, Sweden) because GelRed (Olerup SSP^®^ GelRedTM, Olerup, West Chester, PA, USA, Cat. No. 103.302-05/25) dye molecules are bound to DNA.

Each agarose gel was photographed, marked, analysed, and archived. The presence of alleles was determined by identifying PCR products of size 215 base pairs for GSTM1 and 480 base pairs for GSTT1. The absence of GSTM1 or GSTT1 PCR products indicates a deletion polymorphism or null genotype. Patients with one or two copies of the tested gene were classified as positive genotype (“1”) and homozygous deletions as null genotype (“0”).

A 312-bp fragment of the CYP1A1 gene provides proof of successful PCR; patients with both positive genotypes were used as positive controls.

It was not possible to distinguish homozygous or heterozygous carriers of the positive genotype using this analysis [35].

### 2.4. Statistical Methods

Data were prepared using a Microsoft Office Excel spreadsheet calculator (version 2406,16.0.17726.20078). Data are presented in tables and graphics. The Shapiro–Wilk test was used to analyse the distribution of continuous numerical values. Categorical values are given by their corresponding frequencies and shares.

For comparison of categorical variables between the groups, logistic regression models were fitted with group as dependent variable and categorical variable of interest as the independent variable, along with gender and age at disease onset. The null model was fitted using only gender and age as predictors. For comparison of continuous variables, linear regression models were fitted with continuous variables as dependent variable and group, gender, and age as independent variables. Prior to model fitting, non-normally distributed continuous variables were transformed by inversion transformation of ranks to standard normal distribution to meet linear model assumptions. Log-likelihood test was used to compare the two nested models, and result was used to assess the independence of variables of interest and the group.

*p* values of less than 0.05 were considered significant. We used R version 4.4.0 [36] and the following R packages: gt v. 0.10.1 [37]. 2024), gtsummary v. 1.7.2 [38], openxlsx v. 4.2.5.2 [39], tidyverse v. 2.0.0 [40]. 

## 3. Results

### 3.1. Study Participants’ Characteristics

The study included 109 patients with JIA (35 boys and 74 girls, respectively) diagnosed with oligoarticular, polyarticular, and systemic types of JIA in 46, 56, and 7, respectively. 

All patients were divided into two groups: 46/109 (42.2%) patients who achieved stable remission during MTX treatment for a minimum of 6 months (“MTX” group) and 63/109 (57.8%) patients who did not achieve stable remission during MTX treatment, i.e., patients in whom stable clinical remission was achieved by adding bDMARD to MTX or replacing MTX with bDMARD (“MTX + bDMARD” group) (Table 1). 

Differences in clinical and laboratory categorical variables regarding therapy and disease remission between the study groups are shown in Table 2. Significant differences were detected in wrists, elbows, and temporomandibular joint involvement and uveitis occurrence.

Differences in the observed continuous clinical and laboratory values between the study groups regarding the achieved remission during MTX treatment are presented in Table 3.

In all patients with JIA who did not achieve remission during MTX treatment, remission was achieved by adding bDMARD to MTX or replacing MTX with bDMARD. Clinical remission was achieved in 49/63 (77.8%) patients with the first bDMARD and in 14/63 (22.2%) patients after the replacement of one or more bDMARDs.

### 3.2. Polymorphisms in GST Genes

Table 4 presents the differences in GSTM1 and GSTT1 deletion polymorphisms between the study groups with respect to the remission achieved during MTX treatment.

No significant differences were observed in the distribution of individual deletion polymorphisms or their combinations between the study groups.

## 4. Discussion

According to contemporary therapeutic protocols, MTX is the first drug of the second line in JIA treatment [6,7,8,9]. In view of the significant proportion of patients with JIA in whom MTX does not have an effect even after long-term administration, there is growing interest in finding a marker that can predict the probability of a favourable therapeutic response [15,16,17,18]. Previous attempts to identify a valid clinical, biochemical, molecular, or genetic marker for the prediction of MTX efficacy in JIA have failed or still require validation in large patient cohorts [16,17,18,41,42,43,44,45,46,47].

Although the standard pathway of metabolism and elimination of MTX is well known, previous studies have not been able to unambiguously connect specific gene polymorphisms with changes in MTX-metabolising enzyme activity, i.e., the efficacy of MTX included in therapeutic protocols for various diseases, including JIA [17,46,47,48,49]. Additionally, concomitant folate supplementation, intended to reduce MTX-related side effects, does not weaken its anti-inflammatory effects in patients with RA and suggests that other mechanisms of action and metabolism should also be considered [50].

Because deletion polymorphisms of a specific GST gene terminate the enzymatic activity of a specific GST enzyme, in addition to the possible potentiation of the toxic effects of certain xenobiotics, better therapeutic effects in certain diseases are expected due to their slow elimination and longer bioavailability [24,51,52].

Our study aimed to explore the possible predictive role of GSTM1 and GSTT1 polymorphisms in MTX responsiveness in JIA patients, hypothesising that the deletion variants resulting in decreased enzymatic function boost MTX efficacy in disease control.

GSTM1 deletion polymorphism was detected in 27/46 and 40/63 (58.7% vs. 63.5%, *p* = 0.567), GSTT1 deletion polymorphism in 8/46 and 14/63 (17.4% vs. 22.2%, *p* = 0.502), and a combination of GSTM1 and GSTT1 deletion polymorphisms in 6/46 and 8/63 (13% vs. 12.7%, *p* = 0.966) of our patients with JIA in whom stable clinical remission lasting for at least 6 months was achieved during MTX therapy and patients with JIA in whom stable clinical remission was not achieved during MTX treatment, respectively. Therefore, no significant differences were observed in the distribution of individual GST deletion polymorphisms or their combinations between the study groups.

The obtained results are not consistent with the hypothesis of this study.

Our findings regarding GSTM1 and GSTT1 deletion polymorphisms are most closely consistent with those reported by Kim et al. In their investigation of the pharmacogenetic impact of MTX resistance and toxicity in patients with ALL, the authors did not identify any link between GSTM1 and GSTT1 deletion polymorphisms and MTX toxicity or resistance, despite GSTs being implicated in MTX metabolism. However, they strongly advocate for more research in this area because the significant interpatient variability in drug responses is attributed to numerous germline polymorphisms in genes encoding proteins relevant to pharmacokinetics and pharmacodynamics and because defining interpatient differences in drug pharmacokinetics could improve individualised therapy, potentially overcoming drug resistance and enhancing disease outcomes [30].

From another perspective, the results vary.

According to research by Kalyan Kolli et al. [23], MTX induces an increase in GSTs activity in the small intestines of treated rats, suggesting that at least part of MTX is metabolised by GSTs. Vice versa, a change in GST activity should affect MTX accumulation. Our research contradicts the stated assumption. Animal models are not ideal, and the obtained results cannot be copied unequivocally to humans in all cases. Additionally, GSTs are tissue-specific, and locally induced activity in the small intestine differs from that in other tissues.

A study by Audemard-Verger et al. [24] explored the relationship between GST deletion polymorphism and increased nephrotoxicity due to the possible accumulation of enzyme substrates in patients with SLE treated with cyclophosphamide (CYC). Although the relationship between the GSTM1 null genotype and CYC nephrotoxicity in patients with SLE was confirmed with the subsequent assumption of a similar influence of the same mechanisms on the CYC efficiency, we did not prove the enhanced MTX efficiency in JIA patients through the same presumed mechanism.

Our results also differ from those reported by authors who investigated correlations between GST polymorphisms and clinical responses in different therapeutic modalities, including MTX-containing protocols, in patients with SLE, IBD, and malignant diseases [25,26,27,28,29,31,32,33]. According to the conclusions of these studies, the positive modulatory effects of GST deletion polymorphisms on the efficacy of MTX-containing therapeutic protocols must be considered from the perspective of the potential effects of other xenobiotics used in combination with MTX for treating specific diseases [23,27,32,33]. Our findings indicate that negative modulation effects are also applicable to MTX-containing protocols for JIA treatment.

Exploring less conventional GST polymorphisms, as suggested by some authors, could potentially lead to distinct investigation outcomes [20,53,54,55,56].

An additional noteworthy result obtained from the evaluation of our patients’ demographic, clinical, and laboratory characteristics was the negative predictive value of uveitis in terms of MTX efficacy in patients with JIA. Although MTX has a consistent role in JIA uveitis therapy [9], to the best of our knowledge, there is no literature evidence linking uveitis to MTX effectiveness prediction. Because uveitis is the most serious complication of JIA and requires more frequent bDMARD treatment than arthritis for remission [48], our result is unsurprising. Other findings of our additional evaluation mainly differed from those of other authors [15,16,42,43,49], most likely due to differences in patient populations. Additionally, according to some authors, incorporating individual variables into predictive models can enhance their predictive value [16].

In addition to its relatively small sample size among the Croatian population, the primary constraint of our study was genotyping, which was limited to the detection of complete absence vs. the presence of one or two alleles in GSTM1 and GSTT1. Molecular techniques now capable of distinguishing all three GST genotypes (homozygous presence: 1/1, heterozygous: 1/0 or 0/1, and homozygous deletion: 0/0) in comprehensive multicentric cohort studies on different populations might yield contrasting findings related to the association between GSTT1, GSTM1, and other GTS genetic variants and the effectiveness of MTX in patients with JIA.

## 5. Conclusions

According to our study, no positive correlation was detected between the therapeutic response to MTX and the GSTM1 and GSTT1 deletion polymorphisms. Thus, isolated determination of GSTM1 and GSTT1 deletion polymorphisms or their combination is not suitable for predicting MTX efficacy in JIA.

Nevertheless, because contemporary studies suggest using diverse prediction models composed of numerous well-known and newly discovered clinical and laboratory (including pharmacogenetic) determinants and subjecting them to multifaceted assessments of prognostic power, further studies on GST polymorphisms and their combinations in extensive JIA patient populations hold significance for further research.

## Figures and Tables

**Table 1 biomedicines-12-01642-t001:** Patients according to gender and age at disease onset.

Characteristic	Male			Female		
	MTXN = 17 ^1^	MTX + bDMARDN = 18 ^1^	*p* Value ^2^	MTXN = 29 ^1^	MTX + bDMARDN = 45 ^1^	*p* Value ^2^
Age at disease onset (mos)	73 (48, 115)	83 (28, 113)	0.729	97 (54, 165)	64 (23, 129)	0.049

^1^ Median (IQR), ^2^ Wilcoxon rank sum test; MTX, methotrexate; bDMARD, biological disease-modifying antirheumatic drug; mos, months.

**Table 2 biomedicines-12-01642-t002:** Differences in clinical and laboratory categorical variables between the study groups regarding therapy and disease remission.

Characteristic		MTX, N = 46 ^1^	MTX + bDMARD, N = 63 ^1^	*p* Value ^2^
JIA type	Oligoarticular	20 (43.5%)	26 (41.3%)	0.608
	Polyarticular	23 (50.0%)	33 (49.2%)	
	Systemic	3 (6.5%)	4 (6.3%)	
ANA	Positive	29 (63.0%)	30 (47.6%)	0.353
	Negative	17 (370%)	33 (52.4%)	
RF	Positive	43 (93.5%)	61 (96.8%)	0.665
	Negative	3 (6.5%)	2 (3.2%)	
MTP + IP joints	No	38 (82.6%)	54 (85.7%)	0.754
	Yes	8 (17.4%)	9 (14.3%)	
Ankles	No	28 (60.9%)	42 (66.7%)	0.272
	Yes	18 (39.1%)	21 (33.3%)	
Knees	No	18 (39.1%)	15 (23.8%)	0.162
	Yes	28 (60.9%)	48 (76.2%)	
Hips	No	40 (87.0%)	54 (85.7%)	0.404
	Yes	6 (13.0%)	9 (14.3%)	
MCP + IP joints	No	28 (60.9%)	43 (68.3%)	0.717
	Yes	18 (39.1%)	20 (31.7%)	
Wrists	No	41 (89.1%)	45 (71.0%)	0.011
	Yes	5 (10.9%)	18 (29.0%)	
Elbows	No	44 (95.7%)	54 (85.7%)	0.055
	Yes	2 (4.3%)	9 (14.3%)	
TM joints	No	46 (100%)	60 (95.2%)	0.044
	Yes	0 (0%)	3 (4.8%)	
Uveitis	No	44 (95.7%)	50 (79.4%)	0.030
	Yes	2 (4.3%)	13 (20.6%)	

^1^ n (%); ^2^ Fisher’s exact test; MTX, methotrexate; bDMARD, biological disease-modifying antirheumatic drug; JIA, juvenile idiopathic arthritis; ANA, antinuclear antibodies; RF, rheumatoid factor; MTP, metatarsophalangeal; IP, interphalangeal; MCP, metacarpophalangeal; TM, temporomandibular.

**Table 3 biomedicines-12-01642-t003:** Differences in the observed continuous clinical and laboratory values between the study groups with respect to the remission achieved after methotrexate therapy or the additional introduction of biological therapy.

Characteristic	MTX, N = 46 ^1^	MTX + bDMARD, N = 63 ^1^	*p* Value ^2^
Time span between disease onset and MTX introduction (mos)	3 (1, 14)	4 (2, 9)	0.751
ESR (mm/h) (at disease onset)	25 (15, 40)	22 (14, 33)	0.344
CRP (g/L) (at disease onset)	4 (2, 13)	5 (2, 10)	0.960
HTC (L/L) (at disease onset)	0.34 ± 0.03	0.33 ± 0.03	0.608
L (×10⁹/L) (at disease onset)	9.30 (7.95, 11.50)	9.40 (8.10, 11.25)	0.545
PLT (×10⁹/L) (at disease onset)	386 (326, 438)	401 (352, 456)	0.858
ESR (mm/h) (at MTX introduction)	20 (8, 44)	20 (13, 31)	0.598
CRP (g/L) (at MTX introduction)	3 (1, 11)	4 (2, 11)	0.601
HTC (L/L) (at MTX introduction)	0.35 ± 0.03	0.34 ± 0.03	0.841
L (×10⁹/L) (at MTX introduction)	9.40 (7.00, 11.20)	9.70 (7.85, 11.50)	0.588
PLT (×10⁹/L) (at MTX introduction)	370 (289, 426)	389 (318, 424)	0.689

^1^ Median (IQR); Mean ± SD; ^2^ Wilcoxon rank-sum test; Welch Two Sample *t*-test; bDMARD, biological disease-modifying antirheumatic drug; CRP, C reactive protein; HTC, haematocrit; L, leucocytes; MTX, methotrexate; ESR, erythrocyte sedimentation rate; PLT, thrombocytes.

**Table 4 biomedicines-12-01642-t004:** Differences in GSTM1 and GSTT1 deletion polymorphisms between the study groups.

Characteristic		MTX, N = 46 ^1^	MTX + bDMARD, N = 63 ^1^	*p* Value ^2^
GSTM1	Deletion polymorphism (0)	27 (58.7%)	40 (63.5%)	0.567
	Without deletion polymorphism (1)	19 (41.3%)	23 (36.5%)	
GSTT1	Deletion polymorphism (0)	8 (17.4%)	14 (22.2%)	0.502
	Without deletion polymorphism (1)	38 (82.6%)	49 (77.8%)	
GSTM1/GSTT1	(0/0)	6 (13%)	8 (12.7%)	0.966
	(0/1, 1/0, 1/1)	21 (45.7%)	32 (50.8%)	

^1^ n (%); ^2^ Fisher’s exact test; bDMARD, biological disease-modifying antirheumatic drug; MTX, methotrexate; GSTM1, glutathione S-transferase M1; GSTT1, glutathione S-transferase T1.

## Data Availability

Additional data are available from the corresponding author upon reasonable request.

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
