# Peer review of "Glutathione S-Transferase Gene Polymorphisms as Predictors of Methotrexate Efficacy in Juvenile Idiopathic Arthritis"

_biomedicines, 2024, doi:10.3390/biomedicines12081642_

Round 1

Reviewer 1 Report

Comments and Suggestions for Authors

Reference: Manuscript ID:  biomedicines-2916537

 It is highlighted in blue and italics, the parts of the text that are exposed in a literal way.

The paper titledGlutathione S-transferase gene polymorphisms as predictors of methotrexate efficacy in juvenile idiopathic arthritis, addresses an interesting topic.

On lines 117-180, the authors show what they do in this article, as follows:

The aim of this study was to investigate the possible effects of GSTM1 and GSTT1

polymorphisms on MTX efficacy. We presumed that null GST polymorphisms with a consequent lack of enzymatic activity will result in MTX accumulation and thus increased efficacy in patients with JIA.

However, the data analysis is not conducted rigorously, and the authors do not meet their proposed objectives. Non-experimental research has been carried out with a causal-shared purpose (the authors have not defined it, but they must do so); however, the lack of statistical control and the lack of control in the data analysis is notable. Therefore, the data analysis must be completely redone. Below, I refer to some aspects that support what I say:

1.-The authors write in lines 100-102:

Gender, age at disease onset, disease duration before the initiation of MTX therapy, type of JIA, and body surface area appear to be irrelevant in predicting MTX efficacy [14,40].

However, in Tables 1 and 2, it can be seen that men are in a more significant proportion in MTX and women in MTX+. The authors should check using a Z test whether the percentages in each of the two rows of Table 1 are equal or not. Although p=.409 as a result of the Fischer-Freeman-Halton test, it is possible that the Z test provided lower p values.

On the other hand, in Table 2, they conclude that there are no statistically significant differences (p=.073). I, however, see something else. Although the minimum and maximum values are similar in the two groups, the distribution of the age variable is very different in both groups because the values P25, P50, and P75 are very different. I would look to see if there are any atypical scores concerning age in any group. I don't know what's happening, but age, from my point of view, is “marginally not statistically significant”, and this variable should be controlled statistically. Furthermore, there may even be an interaction between sex and age. In no case have the authors considered the possibility of any significant interaction.

2.- Tables 2, 4, and 5 must be redone. There cannot be frequencies less than 5 in any box. No statistical test is robust to this aspect.

3.- Table 3, from my point of view, should control age and sex, and both variables should be introduced in the data analysis. If the skewness and kurtosis are not greater than 4, they could use robust parametric statistics to control for heterogeneity, introduce more variables into the model (sex as an assigned independent variable), and age as a covariate. They should use multivariate tests (to control Type I error). Even if they finally decide to use non-parametric tests, they should expose the variables' means and variances. These analyses (done well) are helpful, but they are a mere initial approximation to carrying out the objective of the research.

4.- They should select the most critical variables that allow them to differentiate between both groups and use some statistical classification technique (some cluster analysis) to see if they have the power to distinguish between the two groups of subjects.

Comments on the Quality of English Language

Minor editing of English language required

Reviewer 2 Report

Comments and Suggestions for Authors

- “Juvenile idiopathic arthritis (JIA) is the most common rheumatic disease of childhood”. Probably, the authors should specify among chronic rheumatic disorders. Indeed, HSP could be included in the rheumatic disorders and is more common in general.

- In the introduction, the authors should spend some more words on the 1st/2nd line bDMARDs used in JIA, especially in Europe, also better highlighting that MTX intolerance/unresponsiveness is a main criterion to consider biologics (refer to: Biologics for the Treatment of Juvenile Idiopathic Arthritis. Curr Med Chem. 2018;25(42):5860-5893. doi: 10.2174/0929867325666180522085716)

- Said that, I think the length of the introduction is really excessive as well as the high number of references used here. Some may be useful later for the discussion. Indeed, the introduction should provide a general and clear background. The most important detailed should be given in the discussion.

- Conversely, patients and methods section is not well organized and should provide much more details. First of all, I recommend to organize it in subsections (study design and population, ethical statement, laboratory analyses, data collections, statistical methods, others).

- For instance, the number of study participants is not a methodological aspect, but a result, deriving from clear inclusion and exclusion criteria, which should be clearly listed.

- IRB approval number and date should be disclosed. Moreover, the informed consent procedure should be clarified, since legal guardians should give it, along with pediatric patients’ assent, according to the international standard.

- What about the study design and period?

- Results: a first subsection entitled study participants’ characteristics should be added. It should include all the demographic, clinical and laboratory characteristics.

- The organization of the results and its analysis is unclear to me to some extent. Is it possible to analyze patients in groups related to the genetic status?

- I am available to review the discussion, but after the authors clarify and properly revise all the other sections and , especially, the results. At first glance, I think the authors should also reorganize this section. First of all, they should list the main findings at the beginning, avoiding duplication of the introduction. Then, these points should be individually discussed in light of the current literature. Moreover, I think the authors could also mention the relevance of these genetic polymorphisms in the response to methotrexate in other pediatric disease where this drug is used (e.g. ALL, Pharmacogenetic analysis of pediatric patients with acute lymphoblastic leukemia: a possible association between survival rate and ITPA polymorphism. PLoS One. 2012;7(9):e45558. doi: 10.1371/journal.pone.0045558)

Comments on the Quality of English Language

see above

Author Response

Comments and Suggestions for Authors

- “Juvenile idiopathic arthritis (JIA) is the most common rheumatic disease of childhood”. Probably, the authors should specify among chronic rheumatic disorders. Indeed, HSP could be included in the rheumatic disorders and is more common in general.

Thank you for your suggestion. Accepted.

- In the introduction, the authors should spend some more words on the 1st/2nd line bDMARDs used in JIA, especially in Europe, also better highlighting that MTX intolerance/unresponsiveness is a main criterion to consider biologics (refer to: Biologics for the Treatment of Juvenile Idiopathic Arthritis. Curr Med Chem. 2018;25(42):5860-5893. doi: 10.2174/0929867325666180522085716)

The suggested changes have been acknowledged and implemented, referencing the suggested resource.

 Said that, I think the length of the introduction is really excessive as well as the high number of references used here. Some may be useful later for the discussion. Indeed, the introduction should provide a general and clear background. The most important detailed should be given in the discussion.

The suggested changes have been integrated into the introduction, making it more concise.

Conversely, patients and methods section is not well organized and should provide much more details. First of all, I recommend to organize it in subsections (study design and population, ethical statement, laboratory analyses, data collections, statistical methods, others).

- For instance, the number of study participants is not a methodological aspect, but a result, deriving from clear inclusion and exclusion criteria, which should be clearly listed.

- IRB approval number and date should be disclosed. Moreover, the informed consent procedure should be clarified, since legal guardians should give it, along with pediatric patients’ assent, according to the international standard.

- What about the study design and period?

The Materials and Methods section has been rearranged according to your suggestions.

- Results: a first subsection entitled study participants’ characteristics should be added. It should include all the demographic, clinical and laboratory characteristics.

Done as suggested.

- The organization of the results and its analysis is unclear to me to some extent. Is it possible to analyze patients in groups related to the genetic status?

We first grouped patients based on their response to MTX, and then conducted genetic analysis. We aimed to discover a genetic marker capable of predicting MTX response and initiating treatment thereafter. Regardless of the outcome, analysing the effects of Methotrexate based on genetic findings would yield identical results.

- I am available to review the discussion, but after the authors clarify and properly revise all the other sections and , especially, the results. At first glance, I think the authors should also reorganize this section. First of all, they should list the main findings at the beginning, avoiding duplication of the introduction. Then, these points should be individually discussed in light of the current literature. Moreover, I think the authors could also mention the relevance of these genetic polymorphisms in the response to methotrexate in other pediatric disease where this drug is used (e.g. ALL, Pharmacogenetic analysis of pediatric patients with acute lymphoblastic leukemia: a possible association between survival rate and ITPA polymorphism. PLoS One. 2012;7(9):e45558. doi: 10.1371/journal.pone.0045558)

The discussion has been reorganised to include the proposed and a several other related references.

Round 2

Reviewer 1 Report

Comments and Suggestions for Authors

Reference: Manuscript ID:  biomedicines-2916537

Second revision.

I have carefully reread the article. The authors have made a significant effort, but it is insufficient.

With few subjects in each group, it is difficult to have a sufficient sample size to have adequate testing power to test for differences between the groups on any of the dependent variables (because there is likely to be a lot of heterogeneity between the groups). It is difficult to control variables adequately when you have few subjects. And it is even more difficult to control interactions between variables.

Therefore, we must look for other alternatives to obtain the greatest possible amount of information with the data we have.

I suggested controlling variables (sex, age) because, apparently, both variables seem to interact. The authors have not tested that interaction. They have not introduced interaction into the model. They have introduced the variables, but they have not introduced the interaction between the variables). In addition, the authors must put the results of the tested models in a table.

I suggest to the authors that they calculate (with G*Power, for example) the sample size they would need based on the variables that have been most relevant or that can explain differences between the two groups. This way, they could test those “incipient” or “possible” hypotheses, which they could advance in future research.

I also suggest that the authors, with these same data, perform an analysis by propensity scores, equalizing the groups in [Gender, age at disease onset, disease duration before the initiation of MTX therapy, type of JIA, and body surface area appear to be irrelevant in predicting], Just for that reason, because other research has concluded that they are irrelevant variables to explain differences. And then, examine in the two groups created by propensity scores if there are differences between the variables you have examined. If they don't do it now, they could raise it as a possibility.

And with respect to the limitations..., do the authors still think that the sample size is the biggest limitation of this research?

That is, the analysis could be done in greater depth. If the result remains the same, great!, what is going to be done!, we have to look for answers elsewhere, but the data analysis will have been done in-depth, and it will have been done with control. And that is essential to contribute to the replicability of the results.

Author Response

I have carefully reread the article. The authors have made a significant effort, but it is insufficient.

With few subjects in each group, it is difficult to have a sufficient sample size to have adequate testing power to test for differences between the groups on any of the dependent variables (because there is likely to be a lot of heterogeneity between the groups). It is difficult to control variables adequately when you have few subjects. And it is even more difficult to control interactions between variables.

Therefore, we must look for other alternatives to obtain the greatest possible amount of information with the data we have.

I suggested controlling variables (sex, age) because, apparently, both variables seem to interact. The authors have not tested that interaction. They have not introduced interaction into the model. They have introduced the variables, but they have not introduced the interaction between the variables). In addition, the authors must put the results of the tested models in a table.

I suggest to the authors that they calculate (with G*Power, for example) the sample size they would need based on the variables that have been most relevant or that can explain differences between the two groups. This way, they could test those “incipient” or “possible” hypotheses, which they could advance in future research.

I also suggest that the authors, with these same data, perform an analysis by propensity scores, equalizing the groups in [Gender, age at disease onset, disease duration before the initiation of MTX therapy, type of JIA, and body surface area appear to be irrelevant in predicting], Just for that reason, because other research has concluded that they are irrelevant variables to explain differences. And then, examine in the two groups created by propensity scores if there are differences between the variables you have examined. If they don't do it now, they could raise it as a possibility.

 And with respect to the limitations..., do the authors still think that the sample size is the biggest limitation of this research?

That is, the analysis could be done in greater depth. If the result remains the same, great!, what is going to be done!, we have to look for answers elsewhere, but the data analysis will have been done in-depth, and it will have been done with control. And that is essential to contribute to the replicability of the results.

We appreciate your thoughtful recommendations. We performed all the tests that you recommended and obtained similar results each time.   

Thank you for your suggestion to calculate the sample size for future research based on the variables that have been most relevant in our study. We recognize the importance of planning future studies effectively and appreciate your input on this matter.

However, we would like to point out that calculating the sample size based on the observed effect sizes or current results from our study essentially amounts to a form of post-hoc power analysis. This approach can be problematic for the following reasons:

Dependence on Observed Effect Sizes: Using the effect sizes observed in the current study to estimate future sample sizes can be misleading. These observed effect sizes are influenced by sampling variability and may not accurately represent the true effect sizes, potentially leading to over- or underestimation of the necessary sample size.

Redundancy with Observed Results: Calculating the sample size using observed data does not provide new insights beyond what is already reflected in the results and their statistical significance. This calculation is essentially expressing the same results in terms of sample size rather than power or effect size. This is the main problem of post-hoc power analysis (Hoenig, J. M., & Heisey, D. M. (2001). The Abuse of Power: The Pervasive Fallacy of Power Calculations for Data Analysis. The American Statistician, 55(1), 19-24.)

Generalization and Robustness: For future research, it is crucial to consider not only the findings from our study but also a broader range of evidence. Prospective sample size calculations should be informed by a combination of theoretical expectations, meta-analyses, and findings from multiple studies to ensure that they are not unduly influenced by any single set of results.

To address these concerns, we recommend that future sample size estimations be based on comprehensive reviews of the literature and theoretical considerations rather than solely on the current study’s results. This approach provides a more reliable foundation for estimating the required sample size and improves the generalizability of future research findings.

We hope this clarifies our position on the matter. 

Reviewer 2 Report

Comments and Suggestions for Authors

-       In their reply to reviewers the authors stated “We first grouped patients based on their response to MTX, and then conducted genetic analysis. We aimed to discover a genetic marker capable of predicting MTX response and initiating treatment thereafter. Regardless of the outcome, analysing the effects of Methotrexate based on genetic findings would yield identical results.” In the tables, they name the groups as “MTX” and “MTX+bDMARD”. This way I understand that all patients with bDMARD (in addition to MTX) were non-responders to MTX alone. I think the authors should clarify this point in the text.

-       The authors should also state the precise study design at the beginning of the methods section.

-       What about patients who developed MTX intolerance? Apparently, there are no cases in this clinical study.

-       The number of patients reported in table 3 is much larger than what stated in the text at the beginning of the results section and compared to tables 1 and 2. Please, can you clarify?

-       The JIA subtypes in study participants should be indicated.

-       Overall, the methods section should be formatted more properly. Also, section and subsection numbers should be added. The statistical analysis description is dispersive and should be clearer. Finally, the exact polymorphism tested in this research should be better clarified. I suggest carefully revising the methods.

-       In the next revision, the authors should highlight all the changes in word review mode.

-       The conclusion should be revised. For instance, “determination of GSTM1 and GSTT1 gene deletion polymorphisms is not suitable for MTX efficacy prediction in JIA.” To my understanding, the conclusion is that there is no difference between these two groups in terms of the analyzed polymorphism. Moreover, as said, the authors should clarify the JIA subtypes and the indication to the bDMARD, as commented earlier.

-       The discussion should be also extensively revised, since it is quite dispersive and without a clear focus, in the current version.

-       The limitations are more than those discussed by the authors. Please, also expand this aspect.

Comments on the Quality of English Language

See above

Author Response

- We appreciate your thoughtful recommendations. The replies to the questions are beneath each question / section.

In their reply to reviewers the authors stated “We first grouped patients based on their response to MTX, and then conducted genetic analysis. We aimed to discover a genetic marker capable of predicting MTX response and initiating treatment thereafter. Regardless of the outcome, analysing the effects of Methotrexate based on genetic findings would yield identical results.” In the tables, they name the groups as “MTX” and “MTX+bDMARD”. This way I understand that all patients with bDMARD (in addition to MTX) were non-responders to MTX alone. I think the authors should clarify this point in the text.

- Done – added

The authors should also state the precise study design at the beginning of the methods section.

- Done - added

What about patients who developed MTX intolerance? Apparently, there are no cases in this clinical study.

- These patients were excluded from the study (Exclusion criteria, line 115)

The number of patients reported in table 3 is much larger than what stated in the text at the beginning of the results section and compared to tables 1 and 2. Please, can you clarify?

- Typing mistake - third number in a line is supposed to be a superscript.

The JIA subtypes in study participants should be indicated.

- Done

Overall, the methods section should be formatted more properly. Also, section and subsection numbers should be added.

- Done

The statistical analysis description is dispersive and should be clearer. Finally, the exact polymorphism tested in this research should be better clarified.

- Deletion polymorphisms - done

I suggest carefully revising the methods. In the next revision, the authors should highlight all the changes in word review mode.

- Due to extensive changes, especially in discussion section, we did not highlight the changes in word review mode

The conclusion should be revised. For instance, “determination of GSTM1 and GSTT1 gene deletion polymorphisms is not suitable for MTX efficacy prediction in JIA.” To my understanding, the conclusion is that there is no difference between these two groups in terms of the analyzed polymorphism. Moreover, as said, the authors should clarify the JIA subtypes and the indication to the bDMARD, as commented earlier. The discussion should be also extensively revised, since it is quite dispersive and without a clear focus, in the current version. The limitations are more than those discussed by the authors. Please, also expand this aspect.

- Done.

Round 3

Reviewer 2 Report

Comments and Suggestions for Authors

The authors significantly improved the manuscript. I would suggest a few more minor corrections, as follows:

- The IRB approval date should be also disclosed.

- The ethical statement subsection should follow the study design and population section

- As already mentioned, the authors should cleatly indicate the study design in the corresponding section immediately. 

- The dates defining the study period should be precisely identified. 

- A paragraph clearly listing inclusion and exclusion criteria should be created, right after the one defining the study design and period. 

- "2.3. Laboratory analyses (Genotyping of GSTM1 and GSTT1 using polymerase chain reaction (PCR)). One subsection title is needed. The subtitle in brackets should be removed and integrated in the text of the subsection.

- Exact technical specifications could be included. Manufacturer numbers should be reported in brackets. 

- The conclusion should be shorter and only deliver the final message, without summarizing again the study content. 

- Conversely, the discuss on the limitations should be expanded, also because I think these go beyond the small sample size and the laboratory approach. 

- I also suggest an extensive and professional language editing, since gross typing and grammar inconsistencies could be present (e.g. "ancles" in table 3). Moreover, more medical definitions of some joints is needed (e.g. feet and palms joints are not proper medical terms). Please, go over all the manuscript in order to eliminate all the language issues.

Comments on the Quality of English Language

See comments above. Language editing is needed.

Author Response

The authors significantly improved the manuscript. I would suggest a few more minor corrections, as follows:

- The IRB approval date should be also disclosed.

Added.

- The ethical statement subsection should follow the study design and population section

Changed as suggested.

- As already mentioned, the authors should cleatly indicate the study design in the corresponding section immediately. 

Prospecitve cohort study. Added.

- The dates defining the study period should be precisely identified. 

Added.

- A paragraph clearly listing inclusion and exclusion criteria should be created, right after the one defining the study design and period. 

Done.

- "2.3. Laboratory analyses (Genotyping of GSTM1 and GSTT1 using polymerase chain reaction (PCR)). One subsection title is needed. The subtitle in brackets should be removed and integrated in the text of the subsection.

Removed/integrated.

- Exact technical specifications could be included. Manufacturer numbers should be reported in brackets. 

Added.

- The conclusion should be shorter and only deliver the final message, without summarizing again the study content. 

Done.

- Conversely, the discuss on the limitations should be expanded, also because I think these go beyond the small sample size and the laboratory approach. 

Done.

- I also suggest an extensive and professional language editing, since gross typing and grammar inconsistencies could be present (e.g. "ancles" in table 3). Moreover, more medical definitions of some joints is needed (e.g. feet and palms joints are not proper medical terms). Please, go over all the manuscript in order to eliminate all the language issues.

Done.